# Filbertone Reduces Senescence in C_2_C_12_ Myotubes Treated with Doxorubicin or H_2_O_2_ through MuRF1 and Myogenin

**DOI:** 10.3390/nu16183177

**Published:** 2024-09-19

**Authors:** Sumin Jung, Byungyong Ahn

**Affiliations:** 1Department of Food Science and Nutrition, University of Ulsan, Ulsan 44610, Republic of Korea; sminliz210@gmail.com; 2Basic-Clinical Convergence Research Institute, University of Ulsan, Ulsan 44610, Republic of Korea

**Keywords:** muscle aging, filbertone, p53, senescence

## Abstract

It has been demonstrated that filbertone, the principal flavor compound of hazelnuts, exhibits preventive effects against hypothalamic inflammation, obesity, neurodegenerative diseases, and muscle lipid accumulation. However, its influence on muscle aging has yet to be elucidated. The objective of this study was to investigate the effects of filbertone on muscle aging in C_2_C_12_ myotubes subjected to senescence induction by either doxorubicin or hydrogen peroxide. To ascertain the mechanisms by which filbertone exerts its effects, we conducted a series of experiments, including Western blot analysis, reverse transcription quantitative polymerase chain reaction (qRT-PCR), and senescence-associated β-galactosidase (SA-β-gal) staining. Filbertone was markedly observed to decrease not only the protein levels of p53 (*p* < 0.01) in senescence-induced skeletal muscle cells, but also the gene expression levels of p21 (*p* < 0.05), a direct target of p53. The expression of muscle-related genes, including myogenin and muscle RING-finger protein-1 (MuRF1), was found to be significantly enhanced in senescent muscle cells following treatment with filbertone (*p* < 0.05). In addition, the number of senescent skeletal muscle cells exhibiting β-galactosidase activity was found to be markedly reduced in the presence of filbertone (*p* < 0.01). Collectively, these findings suggest that filbertone plays a pivotal role in the regulation of muscle aging.

## 1. Introduction

The skeletal muscle, which constitutes approximately 40% of the body weight in humans, displays a notable morphological alteration with age advancement [1]. One of the defining characteristics of muscle aging is the gradual loss of skeletal muscle, which can be referred to as sarcopenia [2]. This decline is primarily attributed to the reduction in the size and number of muscle fibers, which are of great significance for the generation of rapid forces and power output. The decline in muscle mass is observed to commence as early as the fourth decade of life. This decline is then accelerated beyond the age of 75, resulting in a considerable reduction in both muscle strength and overall functional capacity [3,4].

At the cellular level, the process of muscle aging is characterized by the phenomenon of cellular senescence, which is defined as a state of irreversible cell cycle arrest. This phenomenon is triggered by various stressors, including oxidative stress, DNA damage, and telomere shortening [5,6,7]. Senescent muscle cells exhibit profound alterations in phenotype and function, contributing to the pathogenesis of sarcopenia. The p53-p21 and p16(INK4a)-Rb pathways are two key signaling pathways involved in senescence [8,9,10]. These pathways mediate cell cycle arrest and promote the secretion of pro-inflammatory cytokines, chemokines, and matrix metalloproteinases, collectively known as the senescence-associated secretory phenotype (SASP), which impede the optimal functioning of the immune system, thereby precipitating a state of systemic inflammation [11,12]. Thus, the eradication of senescent cells may prove to be an efficacious approach for the attenuation of aging and the development of age-associated disorders.

Hazelnuts (*Corylus avellana* L.) are a rich source of various nutrients, including fatty acids, proteins, carbohydrates, dietary fiber, and essential minerals such as magnesium and calcium [13]. They also contain polyphenols, which have antioxidant properties. It has been reported that consumption by patients with hypercholesterolemia led to a reduction in total cholesterol and triglycerides. This effect is likely due to the monounsaturated fat, dietary fiber, and antioxidants present in hazelnuts, which collectively influence cholesterol levels and promote cardiovascular health [14]. The principal flavoring component of hazelnuts is filbertone (5-methyl-hept-2-en-4-one). Due to its sweet and citrusy aroma, it is often utilized in the field of perfumery. The Flavor and Extract Manufacturers Association (FEMA) has granted it approval as a safe component [15,16].

The physiological action of filbertone, the active compound in hazelnuts, remains underexplored. Research has indicated that filbertone exhibits beneficial effects such as lowering blood glucose levels, reducing body weight and visceral fat, preventing obesity-induced hypothalamic inflammation, ameliorating neurodegenerative diseases, and regulating muscle lipid metabolism [17,18,19,20]. Despite these findings, there is no conclusive evidence that filbertone directly impacts muscle aging or muscle-related factors. In this study, we suggest that filbertone may protect against muscle aging by mitigating stressors and enhancing muscle-related factors in skeletal muscle cells under stress.

## 2. Materials and Methods

### 2.1. Reagents

Filbertone (5-Methyl-2-hepten-4-one, 98%), doxorubicin, hydrogen peroxide (H_2_O_2_), ammonium persulfate, TWEEN20, protease inhibitor, phosphatase inhibitor, thiazolyl blue tetrazolium bromide (MTT), dimethyl surfoxide (DMSO), ethanol, methanol, and skim milk powder were purchased from Sigma-Aldrich (St. Louis, MO, USA). Dulbecco Modified Eagle’s Medium (DMEM), fetal bovine serum (FBS), Penicililin–Streptomycin, horse serum (HS), and DPBS were purchased from Welgene (Gyungsan, Republic of Korea). A β-galctosidase staining kit was obtained from Cell signaling (Danvers, MA, USA). A SuperSignal West Pico PLUS Chemiluminescent Substrate, BCA protein assay kit, Trizol, and RIPA lysis buffer were supplied by Thermo Fisher Scientific (Waltham, MA, USA). TEMED and 30% acrylamide-bis solution were purchased from Bio-rad (Hercules, CA, USA). M-MLV reverse transcriptase, dNTP Mix, a RNAase inhibitor, and a random primer were obtained from Promega (Madison, WI, USA). TB Green^®^ Premix Ex Taq™ II was supplied by Takara (Shiga, Japan).

### 2.2. RNA Isolation and Quantitative RT-PCR (qRT-PCR)

Total RNAs were isolated from cells using Trizol reagent (Invitrogen, Waltham, MA, USA) according to the manufacturer’s instructions. cDNAs were generated using M-MLV reverse transcriptase with 1.0 µg of total RNA according to the manufacturer’s instructions. qRT-PCR reactions were conducted using TB Green Premix Ex Taq II on the Thermal Cycler Dice Real-Time PCR System (TaKaRa Bio Inc., Shiga, Japan) with specific primers for each gene. All qRT-PCR data were analyzed using TP800 software (TaKaRa Bio Inc.). All primer sets are listed in Appendix A. The expression of target mRNAs was normalized by that of *Rplp0* (36B4) as the standard.

### 2.3. Protein Sample Preparation and Western Blotting

Total Proteins were extracted with RIPA buffer containing 1% NP-40, 1 mM EDTA, protease inhibitor, phosphatase inhibitor, and 1 mM PMSF. The protein concentration was quantified using a BCA Protein Assay Kit (Thermo Fisher Scientific). A protein sample with a protein concentration of 20–30 µg was subjected to sodium dodecyl sulfate-polyacrylamide gel electrophoresis (SDS-PAGE) with an 8–10% polyacrylamide gel and subsequently transferred to a nitrocellulose membrane (NC). The NC blots were incubated in a skim milk solution (5% *w*/*v*) for one hour, following which they were hybridized with primary antibodies at 4 °C overnight. The following antibodies were used: anti-p53 and anti-MuRF1 (Cell Signaling Technology, Danvers, MA, USA), anti-myogenin (Ebioscience, San Diego, CA, USA), and anti-β-actin (Sigma-Aldrich). Protein bands were determined using a SuperSignal West Pico PLUS Chemiluminescent Substrate. Protein detection and quantification were performed using Fusion Solo S (Vilber, France). The intensity of bands was normalized relative to that of β-actin.

### 2.4. Cell Viability Measurement

Cell viability was tested using a 3-(4,5-dimethyl-2-thiazolyl)-2,5-diphenyl-2H-tetrazolium bromide (MTT) assay (Thermo Fisher). For the assay, 5 × 10^4^ cells/mL of C_2_C_12_ cells were seeded on 48-well flat plates. The cells were treated with filbertone (25, 50, 100 µM) for 24 h, after which 10 µL of MTT solution was added per well for 1 h of incubation. Then, the supernatant was removed and colored crystals of formazan were dissolved with 200 µL of dissolving solution (DMSO). Optical density (OD) was read on a microplate reader (BioTek Synergy HTX) at 570 nm.

### 2.5. Senescence-Associated β-Galactosidase (SA-β-gal) Staining

The irradiated cells were subjected to senescence-associated β-galactosidase (SA-β-gal) staining. SA-β-gal staining was conducted on senescence-induced C_2_C_12_ myotubes using a Senescence β-Galactosidase Staining Kit (Cat No. #9860; Cell Signaling Technology, Inc., Danvers, MA, USA) according to the manufacturer’s instructions. The density of SA-β-gal-positive cells for each group was quantified using the ImageJ software provided by the National Institutes of Health (NIH). To quantify the number of SA-β-gal-positive cells, the total number of cells and the number of SA-β-gal-positive cells were counted and the ratio of SA-β-gal-positive cells to total cells was calculated.

### 2.6. Statistical Analysis

The statistical comparisons were evaluated using either the Mann–Whitney U test or one-way ANOVA with Tukey’s multiple post hoc test, conducted using GraphPad Prism 8.0 (San Diego, CA, USA). All results are represented as mean ± SD. It was determined that *p*-values of less than 0.05, 0.01, or 0.005, respectively indicate a statistically significant result.

## 3. Results

### 3.1. Establishment of Cellular Senescence Model in C_2_C_12_ Myotubes

In order to study the cellular senescence signaling pathway in skeletal muscle, we generated a senescence-induced cell model by treatment with doxorubicin or hydrogen peroxide. To this end, we monitored the protein levels of p53, which is known as a senescence marker, tumor protein P53, or transformation-related protein 53 (TRP53), under different conditions. We found that the protein levels of p53 rapidly and markedly increase within 6 h after incubation of doxorubicin (8-fold and *p* < 0.005; Figure 1A,B) or hydrogen peroxide (2-fold and *p* < 0.01; Figure 1C,D) in a time-dependent manner. The findings suggest that p53 protein levels are involved in the aging signal of skeletal muscle cells.

### 3.2. Effect of Filbertone in Doxorubicin or Hydrogen Peroxide Treated C_2_C_12_ Myotubes

We next sought to determine whether filbertone controls the protein levels of p53 in senescence-induced skeletal muscle cells. The level of p53 protein induced by doxorubicin was 40% decreased by filbertone (*p* < 0.01) in a dose-dependent manner (Figure 2A,B). Hydrogen peroxide-induced p53 protein levels were dose-dependently reduced (50% and *p* < 0.01) by filbertone (Figure 2C,D). In addition, we tested the effect of cytotoxicity on the treatment of filbertone in C_2_C_12_ myotubes in a dose- or time-dependent fashion. We found that filbertone does not have any cytotoxicity in C_2_C_12_ myotubes (Appendix A). According to previous reports, p21 is known as one of the primary targets of p53 [21,22,23,24]. Hence, we examined whether the expression of p21 is regulated by filbertone in senescence-induced skeletal muscle cells. The results demonstrate that the gene expression of p21 is significantly downregulated by filbertone in senescent muscle cells (Appendix A). Taken together, these results suggest that the senescence signaling pathway is regulated by filbertone in skeletal muscle cells.

### 3.3. The Activity of Senescence-Associated β-Galactosidase (SA-β-gal) Is Modulated by Filbertone

The senescence-associated β-galactosidase (SA-β-gal) staining assay is a well-established technique for the identification of senescent cells [10,11,25,26]. To investigate whether filbertone regulates the signaling pathway of cellular senescence in skeletal muscle cells, the SA-β-gal staining assay was conducted with doxorubicin or hydrogen peroxide in the context of both the presence and absence of filbertone. Our findings demonstrate that filbertone considerably mitigates cellular senescence by diminishing both the number (*p* < 0.01) and percentage (*p* < 0.01) of doxorubicin-induced senescent cells (Figure 3A–C). Additionally, we observed that filbertone substantially alleviates both the number (*p* < 0.01) and proportion (*p* < 0.01) of senescent cells induced by hydrogen peroxide (Figure 3D–F). These results further imply that filbertone influences the senescence signaling pathway in skeletal muscle cells.

### 3.4. Filbertone Regulates an Associated Risk Factor with Muscle Atrophy

Muscle atrophy, particularly in the context of aging, results in substantial reductions in muscle mass, strength, and functional capacity, thereby adversely impacting the quality of life [27]. One of the principal proteins implicated in this process is MuRF1 (muscle RING-finger protein 1), which acts as an E3 ubiquitin ligase and is instrumental in facilitating the degradation of muscle proteins [28,29]. MuRF1 is highly expressed in both cardiac and skeletal muscles and is upregulated in response to conditions that induce muscle atrophy, such as inactivity, denervation, and fasting [30]. Our findings indicated that MuRF1 protein levels are elevated in response to either doxorubicin (1.4-fold and *p* < 0.05) or hydrogen peroxide (1.5-fold and *p* < 0.05), which ultimately results in the development of senescent skeletal muscle cells. Of particular interest is the observation that filbertone effectively reduces MuRF1 protein levels (*p* < 0.05) in senescent muscle cells (Figure 4A,B). Moreover, our results demonstrate that the gene expression of MuRF1 is enhanced by both doxorubicin (2.5-fold and *p* < 0.05) and hydrogen peroxide (1.8-fold and *p* < 0.05), whereas filbertone administration leads to its reduction (*p* < 0.05) in cells under senescence-inducing conditions (Figure 4C,D).

### 3.5. Filbertone Controls a Myogenic Regulatory Factor

Myogenin is a primary myogenic regulatory factor (MRF). It belongs to a family of basic helix-loop-helix (bHLH) transcription factors that play crucial roles in regulating the development of skeletal muscle cells, called myogenesis. Along with other MRFs like MyoD, Myf5, and MRF4, myogenin facilitates orchestrate the differentiation of myoblasts into mature muscle fibers (myotubes) [31,32,33]. The protein level of myogenin were found to be 35% reduced in senescent skeletal muscle cells induced by both doxorubicin (*p* < 0.05) and hydrogen peroxide (*p* < 0.05). Interestingly, the reduction of myogenin was reversed by filbertone (*p* < 0.05) in senescence-induced skeletal muscle cells (Figure 5A,B). Furthermore, it was observed that the mRNA levels of myogenin are downregulated in senescent skeletal muscle cells and recovered by filbertone (Figure 5C,D).

## 4. Discussion

Sarcopenia, often associated with muscle aging, is characterized by a gradual decline in muscle mass and strength as people grow older. While this decline is a well-documented aspect of the aging process, not all older adults experience it to the extent that it meets the clinical criteria for sarcopenia [4,34,35]. The loss of muscle tissue is accompanied by a reduction in muscle function, which results in diminished physical performance, reduced mobility, and an increased risk of falls and fractures. Concurrently, obesity exacerbates these problems by adding excess weight, which places additional strain on weakened muscles and joints. Muscle aging and obesity are interconnected health issues that significantly impact the quality of life, especially in older adults. The combination of sarcopenia and obesity, sometimes referred to as sarcopenic obesity, leads to a vicious cycle of decreased physical activity and further muscle deterioration [36,37,38]. In addition, the factors contributing to muscle aging include hormonal changes, a reduction in physical activity, inadequate nutrition, and chronic inflammation. The impact of muscle aging extends beyond physical impairments, as it is linked to metabolic disorders such as insulin resistance, type 2 diabetes, and cardiovascular diseases [39,40]. Consequently, a multifaceted approach is required to address muscle aging, including regular exercise, adequate nutrient consumption, and strategies to reduce inflammation. In this study, we examined the potential impact of filbertone on the regulation of the muscle aging process in skeletal muscle cells subjected to senescence induction through the administration of either doxorubicin or hydrogen peroxide. p53 plays a pivotal role in inducing and regulating cellular senescence through various mechanisms such as DNA damage and oxidative stress. We found that p53 is significantly elevated via doxorubicin or hydrogen peroxide within 6 h. Filbertone treatment resulted in the reduction of p53 levels in senescence-induced skeletal muscle cells. In addition, filbertone effectively suppressed the levels of p21, a key target of p53. Our observations also revealed that filbertone affects the activity of senescence-associated β-galactosidase (SA-β-gal), a well-established method for identifying cells undergoing senescence. Collectively, these findings suggest that filbertone exerts a significant influence on the aging process of skeletal muscle.

Muscle atrophy can occur due to a variety of reasons, including disuse (such as prolonged bed rest or immobilization), malnutrition, aging, and certain diseases. One of the key mechanisms underlying muscle atrophy involves the regulation of protein synthesis and degradation within muscle cells. MuRF1 (muscle RING-finger protein-1) is a crucial protein in the process of muscle atrophy, particularly in the context of aging and sarcopenia. It is an E3 ubiquitin ligase, which means it plays a role in tagging proteins for degradation by the ubiquitin-proteasome system [27,28,29]. Various signaling pathways regulate the expression of MuRF1, including the NF-κB pathway, FoxO transcription factors, and the myostatin/Smad pathway [30,41,42]. These pathways respond to various atrophy-inducing stimuli, such as inflammation, oxidative stress, and nutrient deprivation. Interestingly, our results reveal that MuRF1 levels are elevated in conditions that induce cellular senescence. Conversely, filbertone administration led to a notable reduction in these MuRF1 levels in senescent skeletal muscle cells. Therefore, the strategy of inhibiting MuRF1 by filbertone is expected to reduce the muscle loss associated with aging and improve the quality of life in older people.

Myogenic regulatory factors (MRFs) such as myogenin, Myf5, MyoD, and MRF4 play crucial roles in muscle development, repair, and regeneration [33,43,44]. Particularly, myogenin plays a significant role in muscle aging by regulating muscle differentiation and repair processes. Additionally, myogenin is crucial for the proper growth of muscle fibers and the maintenance of muscle stem cell (MuSC) homeostasis. In adult muscle, the absence of myogenin leads to a reduced muscle fiber size, fewer nuclei in muscle fibers, and impaired muscle regeneration [45]. It was exciting to find that the levels of myogenin are decreased by senescent stimuli in skeletal muscle cells, while filbertone significantly increases the levels of myogenin under conditions induced by senescence. Hence, addressing the decline in MuRF1 expression and enhancing myogenin expression in aged muscles could be key strategies in combating age-related muscle loss and improving muscle health in the older adults.

The use of filbertone has significant limitations due to the necessity of achieving extremely high concentrations to realize its potential benefits, which would require an impractically large consumption of hazelnuts, leading to risks such as an excessive intake of calories and fats as well as potential allergic reactions. Additionally, the unavailability of filbertone supplements complicates the ability to control and standardize dosages for therapeutic purposes. Moreover, there are significant regulatory and safety concerns associated with the consumption of high levels of filbertone, as its safety profile at such concentrations has not been sufficiently investigated. In addition, it is important to exercise caution when using chemical methods, such as doxorubicin or H_2_O_2_, as a substitute for the aging process, as these methods induce senescence in vitro rather than reflecting true aging processes. The use of chemically induced senescence rather than naturally aged models limits the generalizability of the findings. To strengthen the conclusions, it would be beneficial to link filbertone to aging-related senescence using an in vivo model. These factors collectively suggest the need for further research and development to explore more practical and safe methods to deliver the benefits of filbertone in a way that is both effective and reflective of true aging processes.

## 5. Conclusions

Consequently, all our observations indicate that treatment with filbertone affects the senescence signals involved in the pathways of p53/p21 signaling, muscle atrophy, and regeneration in skeletal muscle. However, further studies will be essential to examine how filbertone is able to control the precise mechanism of these senescence signaling pathways in skeletal muscle. In summary, our results suggest that filbertone could be regarded as a component in the development strategy of functional foods for the alleviation of muscle aging and age-related diseases.

## Figures and Tables

**Figure 1 nutrients-16-03177-f001:**
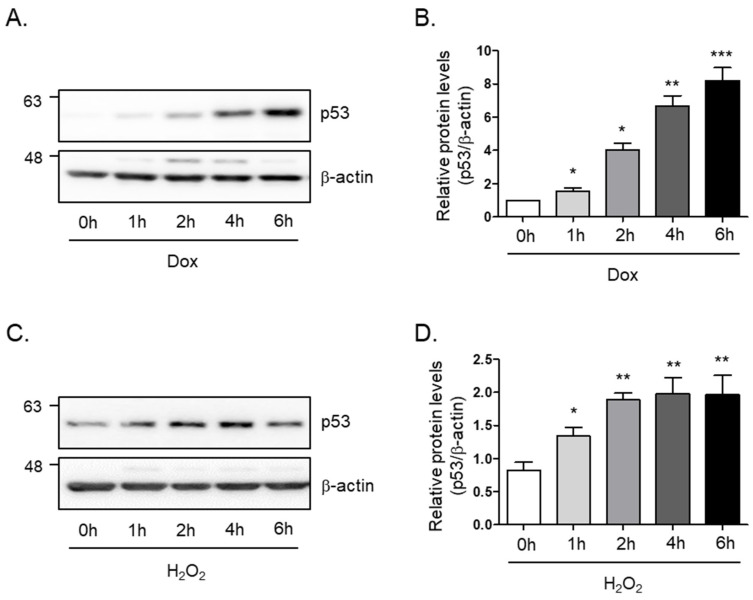
**Doxorubicin or H_2_O_2_ induces cellular senescence in C_2_C_12_ myotubes.** (**A**) Doxorubicin (1 μM) or (**C**) H_2_O_2_ (750 μM) was treated in C_2_C_12_ myotubes for the indicated times. The protein expression of p53 was determined by Western blot analysis. The levels of p53 protein expression were normalized to the β-actin levels in each sample. (**B**,**D**) The graph represents the quantitation of the Western blot data (*n* = 5 per group). All results are expressed as mean ± SD. * *p* < 0.05, ** *p* < 0.01, *** *p* < 0.005 compared to 0 h by one-way ANOVA with Tukey’s multiple comparison post-hoc test.

**Figure 2 nutrients-16-03177-f002:**
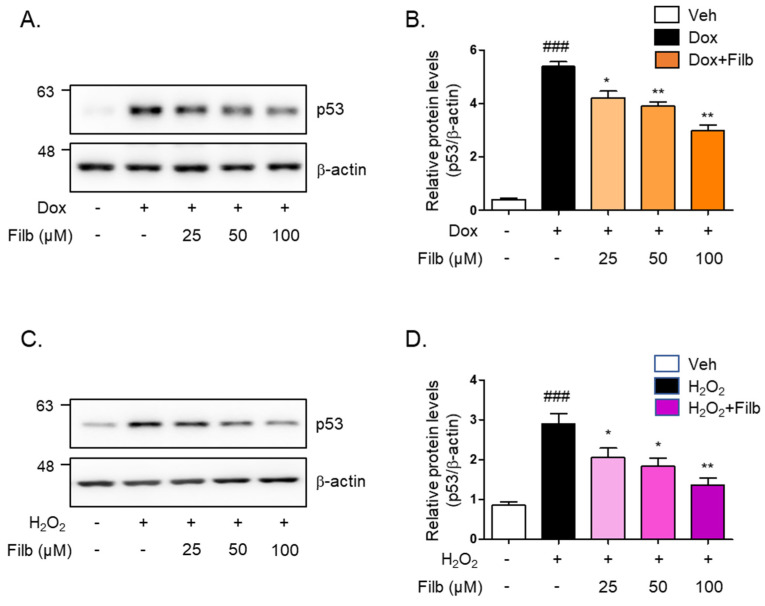
**Cellular senescence is regulated by filbertone.** C_2_C_12_ myotubes were incubated with (**A**,**B**) doxorubicin 1 μM (6 h) or (**C**,**D**) H_2_O_2_ 750 μM (2 h) in the presence of filbertone (25, 50, 100 μM) for 24 h. The protein expression of p53 was determined by Western blot analysis and was normalized to the β-actin level in each sample (*n* = 5 per group). Results are expressed as mean ± SD. ### *p* < 0.005 compared to Veh; * *p* < 0.05, ** *p* < 0.01 compared to Dox (only) or H_2_O_2_ (only) by one-way ANOVA with Tukey’s multiple comparison post-hoc test.

**Figure 3 nutrients-16-03177-f003:**
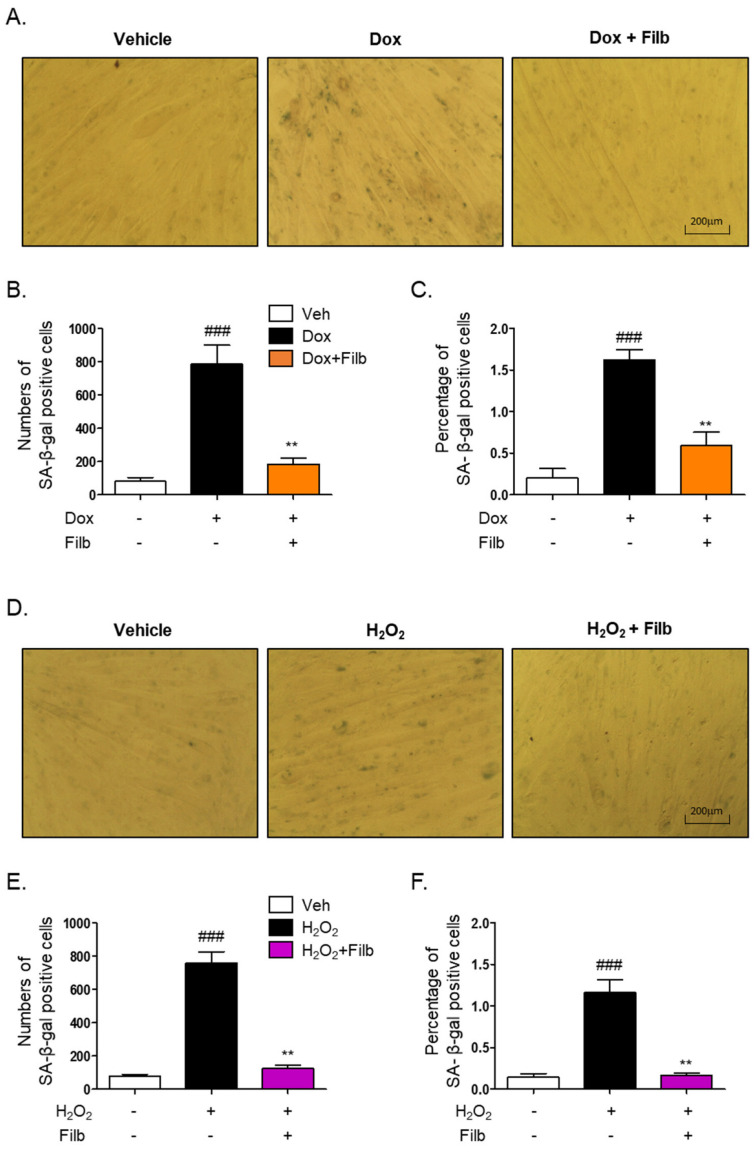
**Effect of filbertone on the senescence-associated β-galactosidase assay in doxorubicin- or H_2_O_2_-treated C_2_C_12_ myotubes.** (**A**) Representative morphologies of senescence-associated β-galactosidase (SA-β-gal) stained (bluish-green color) C_2_C_12_ myotubes with veh, doxorubicin 1 μM (24 h), or doxorubicin 1 μM in the presence of filbertone 100 μM (24 h). (**B**,**E**) SA-β-gal quantification plots based on the count of positive cells (*n* = 3 per group). (**C**,**F**) SA-β-gal quantification plots based on the percentage (%) of area (*n* = 3 per group). (**D**) Representative morphologies of senescence-associated β-galactosidase (SA-β-gal) stained (bluish-green color) C_2_C_12_ myotubes with veh, H_2_O_2_ 750 μM (6 h) or H_2_O_2_ 750 μM in the presence of filbertone 100 μM (24 h). All results are represented as mean ± SD. ### *p* < 0.005 compared to Veh. ** *p* < 0.01 compared to Dox (only) or H_2_O_2_ (only) by one-way ANOVA with Tukey’s multiple comparison post-hoc test.

**Figure 4 nutrients-16-03177-f004:**
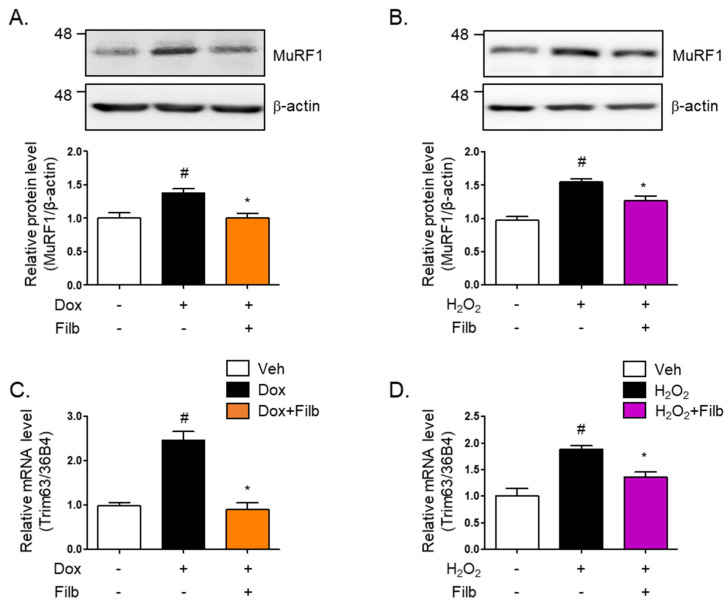
**Effect of filbertone on a risk factor related to muscle atrophy in senescence-induced mouse skeletal muscle.** C_2_C_12_ myotubes were incubated with (**A**) doxorubicin (1 μM for 6 h) or (**B**) H_2_O_2_ (750 μM for 2 h) in the presence of filbertone (100 μM for 24 h). A risk factor related to muscle atrophy (MuRF1) was determined by Western blot analysis and was normalized to the β-actin level in each sample (*n* = 4 per group). (**C**,**D**) Gene expression of trim63(MuRF1) was measured by qRT-PCR and was normalized to the 36b4 level in each sample (*n* = 4 per group). All results are represented as mean ± SD. # *p* < 0.05 compared to Veh; * *p* < 0.05 compared to Dox or H_2_O_2_ by one-way ANOVA with Tukey’s multiple comparison post-hoc test.

**Figure 5 nutrients-16-03177-f005:**
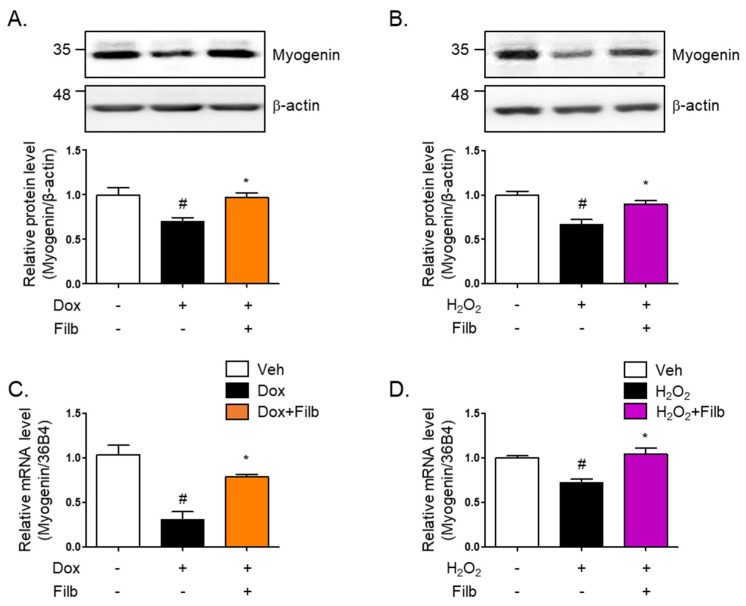
**Effect of filbertone on a myogenic regulatory factor in senescence-induced mouse skeletal muscle.** C_2_C_12_ myotubes were treated with (**A**) doxorubicin (1 μM for 6 h) or (**B**) H_2_O_2_ (750 μM for 2 h) in the presence of filbertone (100 μM for 24 h). A myogenic regulatory factor (Myogenin) was determined by Western blot analysis and was normalized to the β-actin level in each sample (*n* = 4 per group). (**C**,**D**) Gene expression of myogenin was measured by qRT-PCR and was normalized to the 36b4 level in each sample (*n* = 4 per group). All results are represented as mean ± SD. # *p* < 0.05 compared to Veh; * *p* < 0.05 compared to Dox or H_2_O_2_ by one-way ANOVA with Tukey’s multiple comparison post-hoc test.

## Data Availability

The original contributions presented in the study are included in the article, further inquiries can be directed to the corresponding author.

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
