# Peer review of "Filbertone Reduces Senescence in C2C12 Myotubes Treated with Doxorubicin or H2O2 through MuRF1 and Myogenin"

_nutrients, 2024, doi:10.3390/nu16183177_

Round 1

Reviewer 1 Report

Comments and Suggestions for Authors

Overall, I think is a well written piece of science, but it needs some aspects to be confirmed or fixed in order for it to be published.

First of all, the complete images of the nitrocellulose membranes used for Westerns should be submitted as a supplementary zip folder. There has been a rash of researchers manipulating images; to stay one step ahead of any claims of this, please submit the full original images of the blots. Take for instance, the band on the blot presented in the 6th lane of figure 1, panel C corresponding to p53 at 6 hours. That band appears lighter in color than the band to its left corresponding to  p52 at 4 hours, but they have similar values and error bars in panel D.  

Secondly, the title I think is wrong. To me the phrase, "Muscle aging is enhanced by filbertone in senescence-induced C2C12 myotubes" suggests that treatment with flibertone increases the effects of aging on skeletal muscle which is contrary to your findings. I propose "Filbertone reduces senescence in C2C12 myotubes treated with doxorubicin or H2O2. through MuRF1 and Myogenin expression" as the new title.

Thirdly, the y-axis in figure 4 panels A and C incorrectly states "myogenin/beta actin" when it should be "MuRF1/beta actin."

Fourth, lines 231-233 need to be reworded; this is the first sentence of the discussion. Although aging does result in losses in muscle strength and mass which are well documented, not all older adults will lose enough muscle mass and strength to be considered to have sarcopenia. Sarcopenia is a clinical diagnosis for low skeletal muscle strength and quantity with published cut points for Europeans and Asians. 

Fifth, as much as 100 uM filbertone was used in experiments, but this is difficult to contextualize in the context of dietary intake. Please in your discussion, discuss how much filbertone is typically in a serving of hazelenuts. I want to have an idea of many hazelnuts would one need to eat to achieve 100 uM concentration and what implications this has for dietary intake. In other words, would a filbertone supplement be the only viable way of reaching a 100 uM concentration of filbertone?     

Lastly, I would like to add a word of caution when using chemical methods such as doxorubicin or H2O2 as stand ins for aging. Linking flibertone to aging related senescence using an ex-vivo model would really bolster these findings. In other words, I think the work is limited by its in vitro nature and by the fact that the fibers were not truly aging but rather had senescence induced via chemical means. These aspects need to be stated in the discussion as limitations.

Author Response

Responses to Reviewer #1:

We appreciate this Reviewer for an insightful and constructive review. We have taken the comments on board to improve and clarify the manuscript. Please see below. We have included detailed responses to all comments (our replies in blue)

Reviewer # 1

Overall, I think is a well written piece of science, but it needs some aspects to be confirmed or fixed in order for it to be published.

First of all, the complete images of the nitrocellulose membranes used for Westerns should be submitted as a supplementary zip folder. There has been a rash of researchers manipulating images; to stay one step ahead of any claims of this, please submit the full original images of the blots. Take for instance, the band on the blot presented in the 6th lane of figure 1, panel C corresponding to p53 at 6 hours. That band appears lighter in color than the band to its left corresponding to p52 at 4 hours, but they have similar values and error bars in panel D.  

Response: We agree with the reviewer’s recommendations and have revised the manuscript accordingly. All original images for western blots have been included and submitted as a zip folder. We agree with your point that the band of p53 at 6h looks brighter than at 4h. However, all Western blot images in this manuscript were repeated at least 4-5 times so that they have similar values and error bars at 4h and 6h. We think this happened sometimes. That is why the error bars are quite large compared to others and also it has to be repeated more than 3 times.

Secondly, the title I think is wrong. To me the phrase, "Muscle aging is enhanced by filbertone in senescence-induced C2C12 myotubes" suggests that treatment with flibertone increases the effects of aging on skeletal muscle which is contrary to your findings. I propose "Filbertone reduces senescence in C2C12 myotubes treated with doxorubicin or H2O2. through MuRF1 and Myogenin expression" as the new title.

Response: We agree with the reviewer’s point and have updated the manuscript accordingly (line 2-3).

Thirdly, the y-axis in figure 4 panels A and C incorrectly states "myogenin/beta actin" when it should be "MuRF1/beta actin."

Response: We have checked and updated the figure 4 according to reviewer’s comments (line 204).

Fourth, lines 231-233 need to be reworded; this is the first sentence of the discussion. Although aging does result in losses in muscle strength and mass which are well documented, not all older adults will lose enough muscle mass and strength to be considered to have sarcopenia. Sarcopenia is a clinical diagnosis for low skeletal muscle strength and quantity with published cut points for Europeans and Asians. 

Response: We have revised the manuscript according to reviewer’s comments (lines 234-237).

Fifth, as much as 100 uM filbertone was used in experiments, but this is difficult to contextualize in the context of dietary intake. Please in your discussion, discuss how much filbertone is typically in a serving of hazelenuts. I want to have an idea of many hazelnuts would one need to eat to achieve 100 uM concentration and what implications this has for dietary intake. In other words, would a filbertone supplement be the only viable way of reaching a 100 uM concentration of filbertone?     

Response: We agree with the reviewer’s point and have added some explanation in the discussion (line 297-302). It is really hard to reach a 100 μM concentration of filbertone derived from dietary intake. In the literature, we found that maximum filbertone contents in hazelnut is approximately 1,110ug/kg. Mathematically, the requisite quantity of hazelnuts to achieve a filbertone concentration of 100 μM is several kilograms. Thus, as the reviewer mentioned, we think the most effective method for achieving a concentration of 100 μM of filbertone is the administration of a filbertone supplement.

Lastly, I would like to add a word of caution when using chemical methods such as doxorubicin or H2Oas stand ins for aging. Linking flibertone to aging related senescence using an ex-vivo model would really bolster these findings. In other words, I think the work is limited by its in vitro nature and by the fact that the fibers were not truly aging but rather had senescence induced via chemical means. These aspects need to be stated in the discussion as limitations.

Response: We agree with the reviewer’s point and have added the limitation of this study in discussion according to reviewer’s comments (lines 297-312).

Reviewer 2 Report

Comments and Suggestions for Authors

NUTRIENTS-3151501 presents results for filbertone in senescence-induced myotubes. While some parts of the paper were interesting, other areas could be improved. I hope the authors consider my feedback.

·        Abstract: Please insert statistics where appropriate. The results read as narrative without this info.

·        Line 28-30: Sarcopenia or dynapenia might be a better term to explain age-related loss of muscle mass and strength. Line 231 stated sarcopenia.

·        Introduction: Please insert references directly after sentences wherein they are needed throughout. For example, Lines 27-28 need a citation.

·        There needs to be far more details added to the Statistical Analysis section. For example, what specific measures were included in the ANOVAs?

·        Results: Please add relevant statistics to the text (mean differences, p-values, etc.).

·        Line 285: Avoid “elderly” and instead “older adults” throughout.

·        Limitations paragraph needed in the Discussion.

Make any changes to the abstract that align with those made in the text.

Author Response

Responses to Reviewer #2:

We thank this Reviewer for an insightful and constructive review. We have taken the comments on board to improve and clarify the manuscript. Please see below. We have included a point-by-point response to all comments (our replies in blue)

Reviewer # 2

NUTRIENTS-3151501 presents results for filbertone in senescence-induced myotubes. While some parts of the paper were interesting, other areas could be improved. I hope the authors consider my feedback.

  • Abstract: Please insert statistics where appropriate. The results read as narrative without this info.

Response: We have revised the manuscript according to reviewer’s comments (Abstracts)

  • Line 28-30: Sarcopenia or dynapenia might be a better term to explain age-related loss of muscle mass and strength. Line 231 stated sarcopenia.

Response: We agree and have revised the manuscript accordingly (line 30).

  • Introduction: Please insert references directly after sentences wherein they are needed throughout. For example, Lines 27-28 need a citation.

Response: We agree with the reviewer’s point and have updated the manuscript accordingly (line 28, 30, 35).

  • There needs to be far more details added to the Statistical Analysis section. For example, what specific measures were included in the ANOVAs?

Response: We have updated the manuscript according to reviewer’s comments (line 123-127).

  • Results: Please add relevant statistics to the text (mean differences, p-values, etc.).

Response: We agree with the reviewer’s point and have revised the manuscript accordingly (line 135-136, line 148-150, line 173-175, line 197-203, line 220-222). The detailed information also represented in legends.

  • Line 285: Avoid “elderly” and instead “older adults” throughout.

Response: We have revised the manuscript according to reviewer’s comments (lines 290).

  • Limitations paragraph needed in the Discussion.

Response: We agree with the reviewer’s point and have added the limitation of this study in discussion according to reviewer’s comments (lines 297-312).

Round 2

Reviewer 1 Report

Comments and Suggestions for Authors

Thank you for the diligent work on the revisions to the manuscript.

Reviewer 2 Report

Comments and Suggestions for Authors

The authors have addressed my previous concerns.